# Zinc Oxide Nanoparticles and Zinc Sulfate Impact Physiological Parameters and Boosts Lipid Peroxidation in Soil Grown Coriander Plants (*Coriandrum sativum*)

**DOI:** 10.3390/molecules26071998

**Published:** 2021-04-01

**Authors:** Norma Ruiz-Torres, Antonio Flores-Naveda, Enrique Díaz Barriga-Castro, Neymar Camposeco-Montejo, Sonia Ramírez-Barrón, Fernando Borrego-Escalante, Guillermo Niño-Medina, Agustín Hernández-Juárez, Carlos Garza-Alonso, Pablo Rodríguez-Salinas, Josué I. García-López

**Affiliations:** 1Centro de Capacitación y Desarrollo en Tecnología de Semillas, Departamento de Fitomejoramiento, Universidad Autónoma Agraria Antonio Narro, Saltillo C.P. 25315, Coahuila, Mexico; n_nruiz@hotmail.com (N.R.-T.); naveda26@hotmail.com (A.F.-N.); neym_33k@hotmail.com (N.C.-M.); fborregoe9@gmail.com (F.B.-E.); 2Centro de Investigación en Química Aplicada, Blvd. Enrique Reyna Hermosillo, Saltillo C.P. 25294, Coahuila, Mexico; enrique.diazbarriga@ciqa.edu.mx; 3Departamento de Parasitología y Ciencias Básicas, Universidad Autónoma Agraria Antonio Narro, Saltillo C.P. 25315, Coahuila, Mexico; sonia.rmz.barron@gmail.com (S.R.-B.); chinoahj14@hotmail.com (A.H.-J.); 4Facultad de Agronomía, Universidad Autónoma de Nuevo León, Francisco Villa S/N, Col. Ex-Hacienda el Canada, General Escobedo C.P. 66050, Nuevo León, Mexico; guillermo.ninomd@uanl.edu.mx (G.N.-M.); carlos.garza.alonso@gmail.com (C.G.-A.); 5Departamento en Ciencias Biológicas, Facultad de Ciencias Naturales, Universidad Autónoma de Querétaro, Avenida de las Ciencias S/N, Juriquilla, Querétaro C.P. 76230, Juriquilla, Mexico; palanvf@hotmail.com

**Keywords:** nanofertilizer, plant uptake, photosynthetic pigments, lipid peroxidation, enzymatic activity

## Abstract

The objective of this study was to determine the oxidative stress and the physiological and antioxidant responses of coriander plants (*Coriandrum sativum*) grown for 58 days in soil with zinc oxide nanoparticles (ZnO NPs) and zinc sulfate (ZnSO_4_) at concentrations of 0, 100, 200, 300, and 400 mg of Zn/kg of soil. The results revealed that all Zn compounds increased the total chlorophyll content (CHLt) by at least 45%, compared to the control group; however, with 400 mg/kg of ZnSO_4_, chlorophyll accumulation decreased by 34.6%. Zn determination by induction-plasma-coupled atomic emission spectrometry (ICP–AES) showed that Zn absorption in roots and shoots occurred in plants exposed to ZnSO_4_ at all concentrations, which resulted in high levels of hydrogen peroxide (H_2_O_2_) and malondialdehyde (MDA). Only at 400 mg/kg of ZnSO_4_, a 78.6% decrease in the MDA levels was observed. According to the results, the ZnSO_4_ treatments were more effective than the ZnO NPs to increase the antioxidant activity of catalase (CAT), ascorbate peroxidase (APX), and peroxidases (POD). The results corroborate that phytotoxicity was higher in plants subjected to ZnSO_4_ compared to treatments with ZnO NPs, which suggests that the toxicity was due to Zn accumulation in the tissues by absorbing dissolved Zn^++^ ions.

## 1. Introduction

The addition of fertilizers to complement the natural fertility of the soil is essential for crop production, and accurate nutrient management is essential for sustainable agriculture [1,2]. Nanotechnology in agriculture has improved crop production systems with the use of metallic nanoparticles for plant nutrition [3]. Nano-scale fertilizers (1–100 nm) help to improve the efficiency of the use of nutrients by plants due to their small size, greater surface area, and gradual release of their ionic forms [4].

Among metal oxide nanoparticles, zinc oxide nanoparticles (ZnO NPs) have attracted researchers’ attention because of their photocatalytic and photo-oxidant capacity against chemical and biological compounds [5]. In biology and medicine, the possibility of using ZnO NPs has been considered due to their cytostatic activity against cancer cells, the antimicrobial and fungicidal activity because of their high antibacterial efficacy at low concentrations, and their activity against a wide range of strains [6].

However, several studies have shown that ZnO NPs have positive and negative effects on modulating crop growth [7,8]. For example, ZnO NPs application to wheat seeds in a low concentration (10 mg/L) improved water absorption, which resulted in an improved activity of α-amylase and the content of photosynthetic pigments like chlorophyll a and chlorophyll b and the total chlorophyll content [9]. On the contrary, toxic effects of ZnO NPs have been documented in different species. In *Arabidopsis thaliana*, the application of ZnO NPs affected the biosynthesis of chlorophyll and the photosynthetic efficiency by inhibiting the expression of genes associated with chlorophyll synthesis [10].

The effect of ZnO NPs in buckwheat plants (*Fagopyrum esculentum*) at high concentrations (10–2000 mg/L) revealed a significant reduction in biomass accumulation, cell damage in the root surface, and a higher activity of the antioxidant defense system induced by the accumulation of reactive oxygen species (ROS) [11]. ROS are partially reduced forms of atmospheric oxygen like the superoxide radical O_2_•^−^, hydrogen peroxide H_2_O_2_, singlet oxygen O_2_, and hydroxyl radical OH•, which are highly reactive and can cause oxidative stress in organisms [12]. Therefore, malondialdehyde (MDA) concentrations result from lipid peroxidation that is caused by ROS species within the cell [13].

When confronted with oxidative stress generated by ROS, plants activate defense mechanisms, like the antioxidant enzymatic system that operates through the action of a set of enzymes, including ascorbate peroxidase (APX), peroxidase (POD), and catalase (CAT), which play an important role in the protection of cellular components from oxidative damage caused by the absorption of heavy metals in plants [14]. In this way, the effect of the application of ZnO NPs as a plant fertilizer seems to be determined by the applied concentration, the dissolution of its ionic forms, and absorption and transport of the active element and its accumulation in plant tissues [15,16,17]. The high concentration of Zn in plant tissues can influence free radical production and MDA levels [18].

Zinc (Zn) is an essential micronutrient involved in many cellular processes, including auxin biosynthesis, photosynthesis, and protein synthesis [19,20]. Nevertheless, the concentration of Zn in the soil is very low, and it occurs in the form of various types of salts, including sphalerite (ZnFe)S, zincite (ZnO), and smithsonite ZnCO_3_. Furthermore, the absorption of this element by plants is determined by the concentration of carbonates (CaCO_3_) and soil pH, which are the main causes that limit the availability of this micronutrient [2,21]. Zinc oxides (ZnO) and zinc sulfates (ZnSO_4_) are commonly used as fertilizers to improve Zn deficiency in plants; however, their application as fertilizers is limited because of their high solubility in soil and their high cost [22]. Therefore, there is growing interest in improving Zn fertilization methods to enhance their efficiency in crop nutrition.

Several studies have addressed the impact of ZnO NPs on physiological and stress parameters in plants [7,8,23]. However, to verify the feasibility of ZnO NPs as a fertilizer in plant nutrition, it is necessary to carry out simultaneous comparative studies to determine the efficiency of nanofertilizers (ZnO NPs) and conventional fertilizers (ZnSO_4_).

Thus, it is important to determine the relative degrees of importance of the release of ions and its absorption in plant tissues, in addition to physiological and stress impacts, which can compromise plants’ development.

Therefore, the objective of this study was to evaluate the exposure effects of ZnSO_4_ and ZnO NPs in coriander plants at concentrations of 0, 100, 200, 300, and 400 mg of Zn/kg of soil. The accumulation of Zn in the tissues, the content of photosynthetic pigments, the responses of oxidative stress parameters, and the activity of antioxidant enzymes were determined to compare the effects of ZnO NPs with other Zn species (ZnSO_4_) to elucidate mechanisms of toxicity.

## 2. Materials and Methods

### 2.1. Zinc-Based Materials and Coriander Seeds

ZnO NPs were purchased from Nanomaterials Research Inc. (Houston, TX, USA), and ZnSO_4_ (ACS reagent ≥ 99.0% purity) was purchased from Sigma-Aldrich (Saint Louis, MO, USA). Coriander seeds were obtained from the Seed Technology Training and Development Center (CCDTS) of the Universidad Autónoma Agraria Antonio Narro (Saltillo, Coah, Mexico).

#### Characteristics of the ZnO NPs Used in this Experiment

The morphology and microstructure of the samples were examined by conventional and high-resolution transmission electron microscopy (TEM and HRTEM), and selected area electron diffraction (SAED) using a FEI-TITAN 80–300 kV microscope (Fisher Scientific, Hillsboro, OR, USA), operated at an acceleration voltage of 300 kV. TEM and HRTEM micrographs were processed using a Fast Fourier transform software (Digital Micrograph 3.7.0, Gatan Software, Pleasanton, CA, USA).

### 2.2. Preparation of Zinc-Based Solutions/Suspensions and Their Application to the Soil

ZnSO_4_ and ZnO NPs were prepared at concentrations of 100, 200, 300, and 400 mg/kg, based on previous studies conducted by Pullagurala et al. [7]. To ensure uniform dispersion of Zn-based compounds, solutions/suspensions were prepared in DI deionized water and homogenized with an Autoscience AS2060B (Scientific Instrumentation, Bs, As) sonicator for 30 min at 110 volts-3 amps and 50 to 60 Ghz. After sonication, the suspensions of the Zn-based compounds were added and mixed manually with inert soil using a mixture of perlite and vermiculite (ratio 1: 1 substrate volume), an inert material with a neutral pH and no nutrient mineral content [24,25], to obtain final concentrations of 100, 200, 300, and 400 mg of Zn/kg of soil.

### 2.3. Plant Growth Conditions

The experiment was carried out under controlled conditions in a greenhouse, with an average daily temperature of 24.7 °C and a relative humidity of 68%. The coriander seeds were sown in black plastic bags (15.4 × 4.3 × 3.5 inches) containing 450 g of soil modified with the Zn compounds. Five replications were assigned per treatment in a completely randomized design, together with the control (soil without Zn compounds), placing ten coriander seeds in each pot. During the culture (58 days after sowing), the plants were watered with 30 mL per day of a nutrient solution (without Zn), using a commercial formulation of macronutrients (FertiDrip N11-P02-K42) in DI deionized water, with a pH of 6.47 and electrical conductivity of 0.41 µS cm^−1^, in equal quantities for each pot, following the manufacturer’s recommendation (AGROformuladora Delta, Monterrey, NL, Mexico).

### 2.4. Plant Harvest

The harvest was carried out on day 59 after sowing by removing ten plants per treatment, which were used for Zn quantification, and for the determination of photosynthetic pigments and biochemical analyzes (H_2_O_2_, MDA, and enzymes antioxidants). During harvest, the plants were separated into roots and shoots (stems and leaves) and rinsed with tap water to remove the excess soil, then rinsed for 20 s with 0.01 M HNO_3_ and three times again with DI deionized water. Next, the plant tissues were stored at −20 °C until analysis.

### 2.5. Elemental Quantification of Zn by Induction Plasma Coupled Atomic Emission Spectrometry (ICP-AES)

After harvesting, 0.2 g of the plant tissue (roots and shoots) was dried at 60 °C in a Yamato Scientific DX 602C oven (Santa Clara, CA, USA) for 72 h. The resulting material was crushed and subjected to acid digestion in a mixture of perchloric acid and nitric acid [26]. Zn concentrations in the tissues were carried out using the acid digestion extract using an ICP-AES Agilent 725-ES atomic emission induction plasma coupled spectrometer (Santa Clara, CA, USA).

### 2.6. Chlorophyll and Carotenoid Pigment Content

The quantitative evaluation of chlorophyll-a (CHLa), chlorophyll-b (CHLb), and total chlorophyll (CHLt) was carried out by the method of Rajput and Patil [27], while carotenoids were determined following the methodology of Macalacham and Zalik [28], using 1 g of fresh leaf material and homogenized with 10 mL of 80% acetone; then, the extract was centrifuged at 5000 rpm for 5 min. The supernatant was recovered for the analysis, and the absorbance of the extracted solution was measured at 480, 510, 645, and 663 nm. The absorbances were used to determine the pigment concentrations of chlorophylls and carotenoids using the following formulas:CHLa mg g^−1^*FW* = 12.7 (*A*663) − 2.69 (*A*645) × *V*/1000 × *W*
CHLb mg g^−1^*FW* = 22.9 (*A*645) − 4.68 (*A*663) × *V*/1000 × *W*
CHLt mg g^−1^*FW* = 20.2 (*A*645) + 8.02 (*A*663) × *V*/1000 × *W*
Carotenoid mg g^−1^*FW* = 7.6 (*A*480) − 1.49 (*A*510) × *V*/1000 × *W*
where

*A* = Absorbance at specific wavelengths

*V* = Final volume of chlorophyll extract in 80% acetone

*W* = Fresh weight of extracted tissue

### 2.7. H_2_O_2_ and MDA Content

The H_2_O_2_ content in the shoots (stems and leaves) was measured according to the method described by Singh et al. [29]. The plant fresh tissue (0.3 g) was homogenized with 2 mL of 0.1% trichloroacetic acid (TCA). After centrifuging at 7500 rpm for 15 min, 0.5 mL of the supernatant was added to 0.5 mL of phosphate buffer (100 mmol L^−1^, pH 7) and 1 mL of 1 M KI. The concentration of H_2_O_2_ was estimated based on the absorbance of the supernatant at 390 nm based on the following formula: (A390 × V_SN_)/(mg sample/V_TCA_), where V_SN_ is the volume of the supernatant and V_TCA_ corresponds to the volume of TCA used in the extraction. The H_2_O_2_ content was expressed as micrograms per gram of fresh weight (µg g^−1^ FW).

The MDA content in the shoots was determined according to Wang et al. [30]. The plant fresh tissue (0.3 g) was homogenized in 3 mL of 10% TCA, the mixture was subjected to centrifugation for 10 min at 12,000 rpm. Two milliliters of the supernatant was recovered, added to 4 mL of 0.6% thiobarbituric acid (TBA, in 10% TCA), and incubated at 100 °C in a water bath for 15 min. After the incubation, the supernatant (containing MDA) was cooled to room temperature, and absorbance at 450, 532, and 600 nm was measured. Total MDA content was calculated with the following formula: 6.45 (A532 − A600) − 0.56 (A450), and the results are expressed as nanomole per gram of fresh weight (nmol g^−1^ FW). The determination of MDA content and activities of the enzyme antioxidant system were carried out in a Thermo Spectronic BioMate3 spectrophotometer (Rochester, NY, USA).

### 2.8. Antioxidant Enzyme Activity

#### 2.8.1. Enzyme Extraction

Fresh shoot tissue samples were processed to determine peroxidase (POD), ascorbate peroxidase (APX), and catalase (CAT) activity. To obtain a protein extract, 0.2 g of tissue was used and placed in a pre-frozen mortar; then, the tissue was macerated with liquid nitrogen, adding 0.1% polyvinylpyrrolidone (PVP), and then the samples were collected in centrifugation microtubes and 1 mL of extraction buffer (100 mmol L^−1^ phosphate buffer, pH 7, supplemented with 0.1 mmol L^−1^ EDTA) was added to each sample. The mixtures were centrifuged at 1200 rpm for 5 min at 4 °C. The supernatants were stored at −20 °C until analysis.

#### 2.8.2. POD Activity

POD activity was determined according to Kwak et al. [31] using pyrogallol as a substrate. The reaction volume was 3 mL, containing 15 µL of the protein extract, 2.5 mL of phosphate buffer (pH 8, 100 mmol L^−1^), and 320 µL of 5% pyrogallol and reacted with 165 µL of H_2_O_2_ at 0.147 mmol L^−1^ (Sigma Aldrich, St Louis, MO, USA). The reaction started with the addition of H_2_O_2_, and the increase in absorbance was measured at 420 nm every 20 s for 1 min. One unit of POD activity is defined as 1.0 mg of purpurogallin formed in 20 s at pH 6 with an ε = 26.6 mM^−1^ cm^−1^.

#### 2.8.3. APX Activity

APX activity was determined as described by Nakano and Asada [32], by measuring the absorbance decrease at 290 nm due to ascorbate oxidation. The reaction mixture comprised 15 µL of protein extract, 885 µL of phosphate buffer (pH 7, 50 mmol L^−1^), and 50 µL of ascorbic acid (10 mmol L^−1^) and reacted with 50 µL of 10 mmol L^−1^ of H_2_O_2_ (ε = 2.8 mM^−1^ cm^−1^). APX activity was expressed as the amount of protein that produces 1 mmol of oxidized ascorbate per minute.

#### 2.8.4. CAT Activity

H_2_O_2_ decomposition was measured by the decrease in absorbance at 240 nm according to Elavarthi and Martin [33]. The reaction mixture comprised 15 µL of protein extract, 965 µL of phosphate buffer (pH 7, 50 mmol L^−1^) and reacted with 20 µL to 0.5 mmol L^−1^ of H_2_O_2_ in a reaction volume of 1 mL (ε = 0.04 µM^−1^ cm^−1^). Enzyme activity was expressed in units per volume (U_i_ mL^−1^), with 1 U_i_ being the amount of enzyme that transforms one micromole of substrate per minute under the standard conditions mentioned above.

### 2.9. Statistical Analysis

The research study was established under a completely randomized design with nine treatments and five experimental units per treatment, and each experimental unit comprised a plastic bag containing ten coriander plants. The results were reported as mean ± standard deviation, the statistically significant differences between the samples were analyzed with an ANOVA, and the means of the treatments were compared with the Tukey Test (*p* ≤ 0.05) using the statistical package SPSS version 21.0 (SPSS Inc., Chicago, IL, USA).

## 3. Results and Discussion

### 3.1. ZnO NPs Characterization

Figure 1a is a typical TEM image showing the morphology ZnO nanoparticles with irregular shape (quasi-spherics). Figure 1b shows the size distribution of the NPs obtained by measuring more than 300 particles, depicting that 63% the sizes of the NPs are between 60 and 140 nm; therefore, ZnO NPs have a relatively wide size distribution. Figure 1c shows the elemental dispersive spectroscopy (EDS) spectra that confirms that the nanoparticles are constituted by Zn and O approximately in 80 and 20 wt%, respectively, while the C and Cu signals are due to the carbon and copper of the TEM grid.

### 3.2. Zinc Uptake in Root and Shoot

Figure 2 shows the Zn concentration in roots and shoots of plants exposed to compounds based on Zn. Figure 2A shows that the accumulation of Zn increased significantly (*p* ≤ 0.05) in the roots of the plants exposed to the different concentrations of ZnSO_4_ and ZnO NPs. It should be noted that all the treatments with ZnSO_4_ presented the highest absorption of Zn in the roots in a concentration-dependent manner, the highest Zn accumulation was observed in root samples exposed to concentrations of 300 and 400 mg/kg with ZnSO_4_; the obtained increases were higher by 23.2% and 36.2%, respectively, compared to the ZnO NPs treatments at the same concentration (300 and 400 mg/kg).

Zn accumulation in the shoots presented a similar trend with respect to the concentrations observed in the roots (Figure 2B). The findings indicate that the shoots with the highest Zn concentration came from plants subjected to different ZnSO_4_ concentrations; 300 and 400 mg/kg of ZnSO_4_ produced increases of 40.1% and 44.7%, respectively, compared to 300 and 400 mg/kg with ZnO NPs. Based on these results, it is evident that the highest Zn concentration was found in the roots and shoots of the plants treated with ZnSO_4_, which is due to a greater Zn absorption by the roots of the plants from this source as a free cation (Zn^++^) and, consequently, to a higher Zn translocation from the root to the shoots.

The high Zn translocation from root to shoot with the application of ZnSO_4_ (Zn^++^ ion), compared to ZnO NPs, is probably due to the high solubility of ZnSO_4_ (approximately 305.42 mg L^−1^), which is much higher than the solubility of the ZnO NPs (89.81 mg L^−1^), which present slow and gradual availability of dissolved Zn [34,35]. Therefore, by using ZnSO_4_ as a fertilizer source, a greater availability Zn for plant roots is achieved with this source. Similar results have been reported, highlighting that the translocation of free Zn^++^ ions contained in ZnO NPs is lower than their ionic counterparts (ZnSO_4_ and ZnCl_2_) [7,12,36]. Therefore, the effect of ZnO NPs depends on the dissolution of Zn on the root surface and the use of the active mineral available for upward transport in the vascular bundles.

### 3.3. Chlorophyll and Carotenoid Content

The results indicate that the photosynthetic pigments (CHLa, CHLb and CHLt) in coriander leaves increased significantly (*p* ≤ 0.05) in all treatments with the application of Zn compounds (Figure 3). The highest accumulation of CHLa, CHLb, and CHLt was obtained at a concentration of 200 mg/kg using the two Zn sources, ZnSO_4_ and ZnO NPs (Figure 3A–C); with ZnSO_4_ at 200 mg/kg, the accumulation of CHLa increased by 52.1%, CHLb by 44.4%, and CHLt by 49.9%, while the ZnO NPs at the same concentrations showed similar increases of 50.6% for CHLa, 42.4% for CHLb, and 48.3% for CHLt compared to the control, although no significant differences were found between ZnSO_4_ and ZnO NPs in terms of accumulation of chlorophyll and carotenoid content at concentrations of 200 and 300 mg/kg (Figure 3).

On the contrary, at 400 mg/kg of ZnSO_4_, the accumulation of CHLa, CHLb, and CHLt decreased by 33.1%, 38.4%, and 34.6%, respectively, compared to the treatment that presented the highest accumulation when applying this Zn source (200 mg/kg of ZnSO_4_). In Figure 3D, the carotenoid content increased for most of the treatments; however, the ZnSO_4_ at 400 mg/kg produced a reduction in the carotenoid content, which was also statistically significant compared to the different concentrations and sources of Zn (*p* ≤ 0.05). Our results are consistent with those of Pullagurala et al. [7], who mentioned that in coriander leaves, the relative content of chlorophyll and carotenoids increased by 41% and 37% because of the application of Zn compounds (ZnO NPs and ZnCl_2_) to the soil at concentrations of 100 and 200 mg/kg, respectively. According to Samreen et al. [37], the supply of Zn through a hydroponic system in bean plants (*Vigna radiata*) increased the content of chlorophyll, proteins, and minerals.

However, other studies have reported that Zn supply to the soil with ZnSO_4_ and ZnO NPs in high concentrations (400–800 mg/kg) exerted harmful effects on the accumulation of chlorophylls and carotenoids in rice (*Oryza sativa*), bean (*Phaseolus vulgaris*), and tomato plants (*Solanum lycopersicon*) [12,38], due to the toxicity generated by the high absorption of Zn^++^ ions within the plant [15]. In this study, concentrations between 100 to 300 mg/kg of ZnSO_4_ and ZnO NPs caused an increase in photosynthetic pigments and carotenoids; on the contrary, with 400 mg/kg of ZnSO_4_, the accumulation of CLHt and carotenoids decreased (Figure 3), which indicates that the higher concentration of Zn^++^ in the shoots from that source (Figure 2B) could generate toxic effects that affect chlorophyll biosynthesis and damage the photosynthetic system [10]. The intensity of the green color of the leaves is due to the presence of photosynthetic pigments (chlorophyll). In the photosynthesis process, carotenoids help to capture light, but they also have another important function, as they are considered the first line of defense against the accumulation of H_2_O_2_, which is an integral part of ROS that generate oxidative stress due to the absorption of metals heavy on plants [39,40]. Therefore, the increases in chlorophyll and carotenoids in coriander plants exposed to the different concentrations and sources of Zn may be related to the greater increase in H_2_O_2_ (Figure 4). The production of excessive ROS can inhibit photosynthetic processes and, consequently, cause greater absorption of light energy that can be used during photosynthesis [41].

While Zn is an essential microelement for plant growth and is required for chlorophyll biosynthesis because it influences the activity of important enzymes like carbonic anhydrase that contains a Zn atom that catalyzes the hydration of CO_2_, facilitating the diffusion of carbon dioxide to the carboxylation sites in plants [7,42], in high concentrations, it can induce the generation of H_2_O_2_ and lipid peroxidation (Figure 4) and lead to a reduction in the biosynthesis of photosynthetic pigments [12,15].

Chlorophyll content has been classified as a reliable indicator in relation to contamination and toxicity of heavy metals in plants [43]. It has been recognized that the increase in H_2_O_2_ and MDA by the absorption of heavy metals is a byproduct of the oxidation of polyunsaturated lipids caused by ROS species [7,12]. Low concentrations of H_2_O_2_ (10 μm) can inhibit CO_2_ fixation by 50% and, therefore, the accumulation of photosynthetic pigments [44].

Therefore, the balance between ROS production and elimination in chloroplasts is delicate and must be strictly controlled by the activity of antioxidant enzymes and components of secondary metabolites [44,45]. This is perhaps surprising, however, if it is considered that ROS generated within the cell participate in a redox signaling cascade, and then could be related to the maintenance of an optimal level of ROS at the cellular level that allows adequate redox biological reactions and the regulation of the numerous essential processes for plants such as growth and development [46,47].

The data from this study suggest that the higher activity of antioxidant enzymes (CAT, APX, and POD, Figure 5), could participate in the modulation of ROS, since the levels of H_2_O_2_ (Figure 4) in coriander plants are not present in quantities that significantly affect the accumulation of photosynthetic pigments (Figure 3). This indicates that the stress levels generated by the application of Zn in its different sources and concentrations were modulated by the activity of antioxidant enzymes and did not affect the chlorophyll content, which is a common indicator of photosynthetic efficiency and one of the most important determinants of plant growth.

### 3.4. H_2_O_2_ and MDA Content

The content of H_2_O_2_ and MDA in the shoots varied significantly (*p* ≤ 0.05) according to the treatment and the source of Zn (Figure 4). The results indicate that the highest H_2_O_2_ content was in the shoots of the plants subjected to 400 mg/kg of ZnSO_4_ (Figure 4A); the increases obtained exceeded 76.8% and 26.6%, respectively, compared to the control and 400 mg/kg of ZnO NPs. The highest levels of MDA content were observed with 300 mg/kg of ZnSO_4_ (Figure 4B), with increases of 78.6% and 30.3%, compared to the control and 300 mg/kg of ZnO NPs; however, using 400 mg/kg of ZnSO_4_, there was a dramatic decrease of 78.6% in the MDA levels. In this work, the highest exposure of ZnSO_4_ (400 mg/kg) triggered the highest activity of anti-stress enzymes and/or antioxidants (Figure 5), which could potentially decrease lipid peroxidation in shoots of plants exposed to 400 mg/kg with ZnSO_4_.

The accumulation of MDA is a byproduct of oxidative damage to membrane lipids. In coriander plants subjected to different sources and concentrations of Zn, it is indicative of an increase in lipid peroxidation and is in accordance with the observation reported by several authors in different plant species [7,12]. In plant cells, redox homeostasis is developed because of the equilibrium between ROS generation and the functioning of the antioxidant enzymes, where an efficient defense system in plants keeps the proper balance between ROS generation and elimination [48]. A basal level of ROS, which is maintained above the toxic concentration, redox regulations in the cells are essential since they allow them to maintain the balance between the production of ROS and their elimination, which can occur through the activity of antioxidant enzymes and can uphold the usual physiological activities of plants [45]. According to these results, all the Zn-based treatments induced stress in the plants; however, the greater absorption of Zn by the roots and its transport to the shoots with the ZnSO_4_ treatments at the highest concentrations (300 and 400 mg/kg, Figure 2) induced greater stress in plants compared to ZnO NPs treatments, which triggered a greater accumulation of H_2_O_2_ and lipid peroxidation, indicating that a greater oxidative damage is induced by the excessive accumulation of Zn in the plant’s tissues. Various studies have reported that H_2_O_2_ and MDA concentrations in plant tissues increase due to the absorption of heavy metals [29,49]. For example, García-Gómez et al. [12] mentioned that the addition of Zn as ZnO and ZnSO_4_ at concentrations of 100 to 225 mg/kg of soil caused a 65% increase of ROS in bean plant tissues, while other studies indicated that the shoots of coriander plants exposed to 100, 200, and 400 mg ZnO per kg of soil, increased the MDA levels by 142, 200, and 173%, respectively [7].

In this study, lipid peroxidation caused by Zn toxicity was lower with the ZnO NPs, compared to the ZnSO_4_ source, the lower toxicity of ZnO NPs could be, at least in part, because of the slower and more gradual release of the Zn^++^ contained in the ZnO NPs, unlike ZnSO_4_, which is highly soluble [35]. Therefore, when applying ZnSO_4_ to the soil, Zn may be more available due to the formation of soluble Zn organic complexes that are transportable and available for absorption by plant roots [50]. However, this allows high concentrations of ZnSO_4_ as a fertilizer to generate toxicity in plants because of its high accumulation in plant tissues, which results in high levels of H_2_O_2_ and lipid peroxidation [2].

### 3.5. Antioxidant Enzyme Activity

CAT activity depended on the Zn source and the treatment used (Figure 5A). Increases in Zn concentration significantly increased (*p* ≤ 0.05) the CAT activity; however, all the ZnSO_4_ concentrations tested were more effective than ZnO NPs in increasing the CAT activity; the highest activity was observed at 400 mg/kg with ZnSO_4_ (Figure 5A); the obtained increases exceeded by 85.6% and 48.3%, respectively, compared to the control and the ZnO NPs at 400 mg/kg.

The increase in APX and POD activities (Figure 5B,C) was similar to that observed in CAT. According to the results, the treatments with ZnSO_4_ were more effective than the ZnO NPs in increasing the activity of APX and POD; at a concentration of 400 mg/kg ZnSO_4_, the APX activity showed increases of 80.5% and 46.1%, respectively, compared to the control and the ZnO NPs at 400 mg/kg (Figure 5B). The POD activity (Figure 5C) at the highest concentration with ZnSO_4_ (400 mg/kg) showed increases of 83.4% and 44.9% when compared to the control and the ZnO NPs at 400 mg/kg.

In general, all the Zn-based treatments induced a greater activity of antioxidant enzymes in the shoots; however, all the ZnSO_4_ concentrations were more effective to increase the activity of CAT, APX, and POD, this can be attributed to the greater increases observed H_2_O_2_ and MDA (Figure 4), which were generated by the greater accumulation of Zn in the shoots (Figure 2) using ZnSO_4_. Various studies have shown that the effects on antioxidant enzyme activity are associated with an increase in ROS and MDA levels, caused by a high concentration of Zn in plant tissues [12,51].

It has been shown that exposure to plants with different sources of Zn can stimulate the generation of ROS species [23]. A high concentration of ROS stimulates the activity of antioxidant enzymes [14]. Antioxidant enzymes play an important role in protecting cellular components from oxidative damage caused by the absorption of heavy metals in plants [52]. The response of defense reactions against ROS is called hormesis [53], which induces defense mechanisms on two levels. The first is the enzymatic level (short-term reaction), in which the activity of antioxidant enzymes is increased. The second level is long-term adaptation and comprises two sublevels: transcriptional and genomic [6]. Plants have differential responses to oxidative stress conditions, which allow them to develop resistance to heavy metal exposure [45]. In this sense, CAT, APX, and POD are the main enzymes involved in H_2_O_2_ detoxification (produced through the dismutation of O_2_^−•^ in peroxisomes and chloroplasts) to H_2_O and O_2,_ to reduce oxidative damage [54]. In this study, the toxic effects of ZnSO_4_ were greater than those observed with ZnO NPs, so our results suggest that the toxic effects of Zn-based fertilizers depend on the dissolution of the element on the root surface and the use of the active mineral available for its absorption in plant tissues. Therefore, the lower toxicity observed with ZnO NPs supports the proposed use of this compound as a nanofertilizer in the future [7,55].

## 4. Conclusions

The results showed that the tissues of coriander plants subjected to different concentrations of ZnO NPs had a lower Zn absorption compared to their ionic counterparts (ZnSO_4_). Concentrations between 100 and 300 mg/kg of ZnO NPs and ZnSO_4_ produced an increased accumulation of CHLa, CHLb, and CHLt, while 400 mg/kg of ZnSO_4_ decreased the accumulation of photosynthetic pigments. It should be noted that ZnSO_4_ more efficiently induced a greater accumulation of H_2_O_2_ and MDA, compared to the ZnO NPs, which resulted in a greater activity of antioxidant enzymes. According to our results, the toxicity of ZnSO_4_ was greater than that generated by the ZnO NPs, which indicates that the toxicity of ZnSO_4_ corresponded to the effects of dissolved Zn^++^ ions that are mobile and are easily transported by the roots, thus allowing a greater accumulation of Zn in plant tissues, which induced greater stress in plants. Therefore, the physiological impact and the antioxidant responses of plants to the sources of fertilization used in this study (ZnO NPs and ZnSO_4_) could largely depend on the solubility of the compound for the release of the active element (Zn^++^).

## Figures and Tables

**Figure 1 molecules-26-01998-f001:**
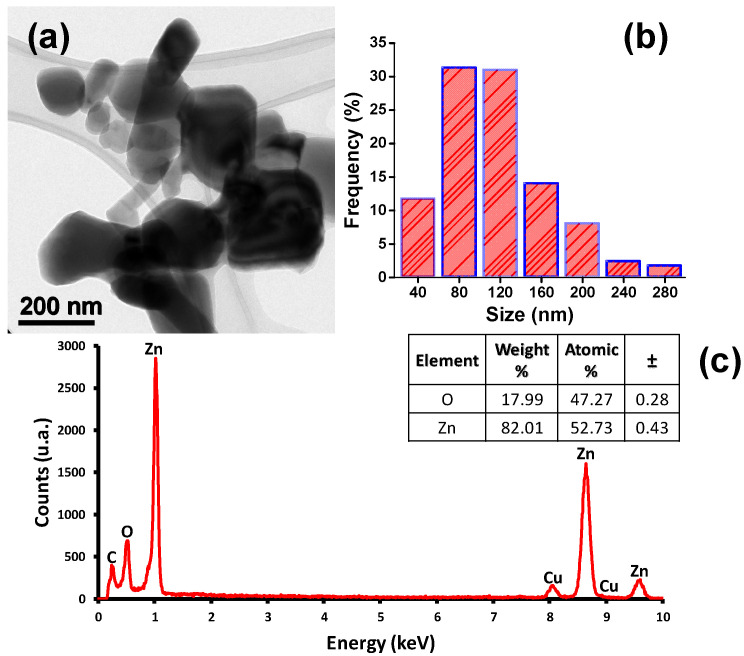
(**a**) The TEM micrograph shows the morphology of the ZnO nanoparticles (NPs) in the sample, (**b**) histogram of the distribution of the NPs sizes, and (**c**) spectrogram of the chemical composition of the sample carried out through elemental analysis (EDS).

**Figure 2 molecules-26-01998-f002:**
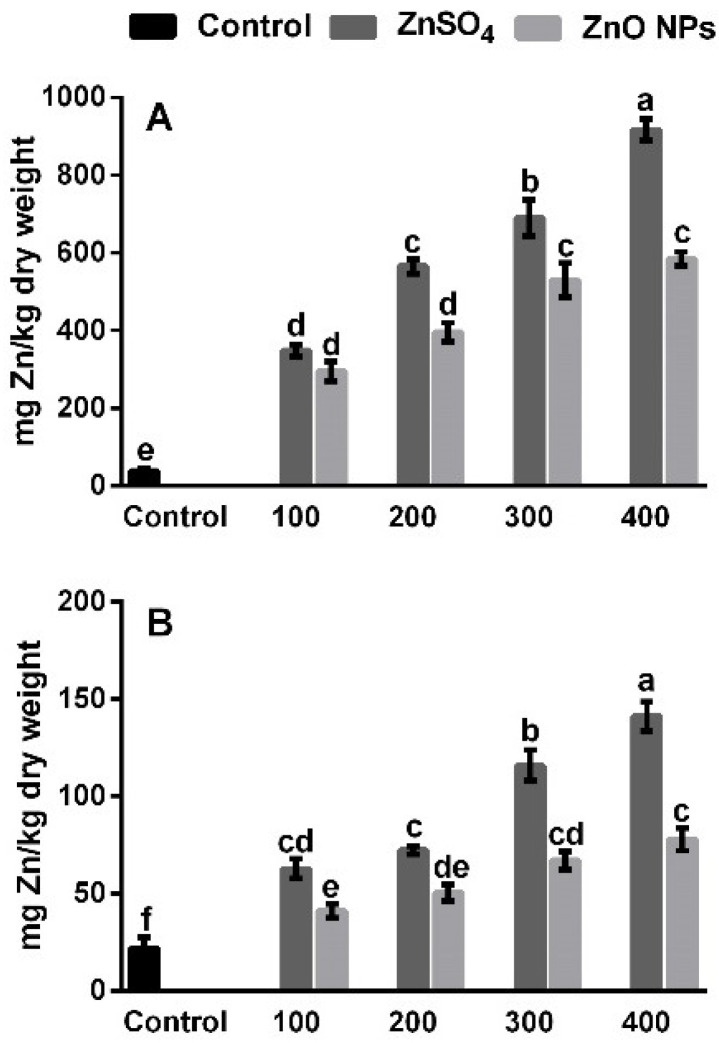
Zn absorption in root (**A**) and shoot (**B**) of coriander plants grown for 58 days in inert soil modified with zinc sulfate (ZnSO_4_) and zinc oxide nanoparticles (ZnO NPs) at concentrations of 0, 100, 200, 300, and 400 mg of Zn/kg of soil. Values are the average of five replications, means (n = 5). Bars represent the standard deviation of the mean. Different letters in each bar mean that the treatments were statistically different (Tukey, *p* ≤ 0.05).

**Figure 3 molecules-26-01998-f003:**
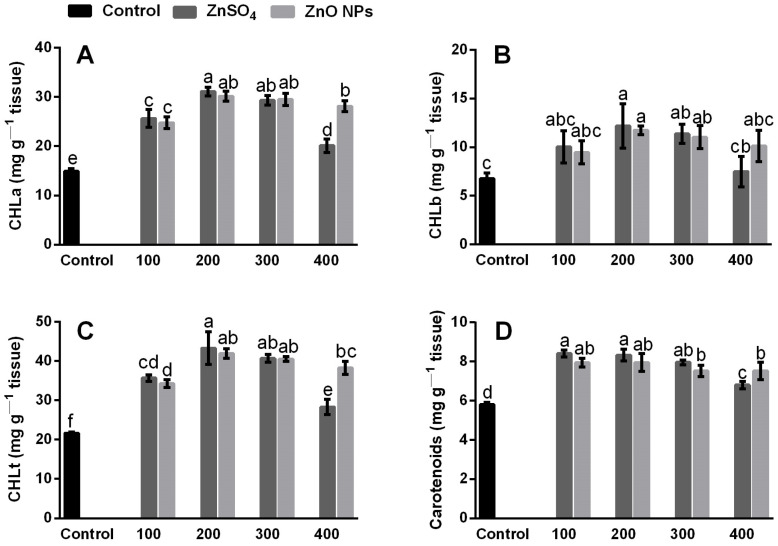
(**A**) Chlorophyll-a (CHLa), (**B**) chlorophyll-b (CHLb), (**C**) total chlorophyll (CHLt), and carotenoids (**D**) of coriander plants grown for 58 days in inert soil modified with zinc sulfate (ZnSO_4_) and zinc oxide nanoparticles (ZnO NPs) at concentrations of 0, 100, 200, 300, and 400 mg of Zn/kg of soil. Values are the average of five replications, means (n = 5). Bars represent the standard deviation of the mean. Different letters in each bar mean that the treatments were statistically different (Tukey, *p* ≤ 0.05).

**Figure 4 molecules-26-01998-f004:**
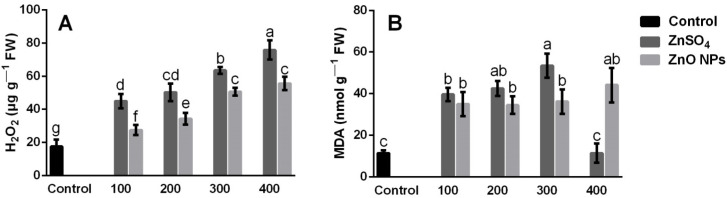
(**A**) Hydrogen peroxide (H_2_O_2_) and (**B**) malondialdehyde (MDA) content in shoots of coriander plants grown for 58 days in inert soil modified with zinc sulfate (ZnSO_4_) and zinc oxide nanoparticles (ZnO NPs) at 0, 100, 200, 300, and 400 mg of Zn/kg of soil. Values are the average of five replications, means (n = 5). Bars represent the standard deviation of the mean. Different letters in each bar mean that the treatments were statistically different (Tukey, *p* ≤ 0.05).

**Figure 5 molecules-26-01998-f005:**
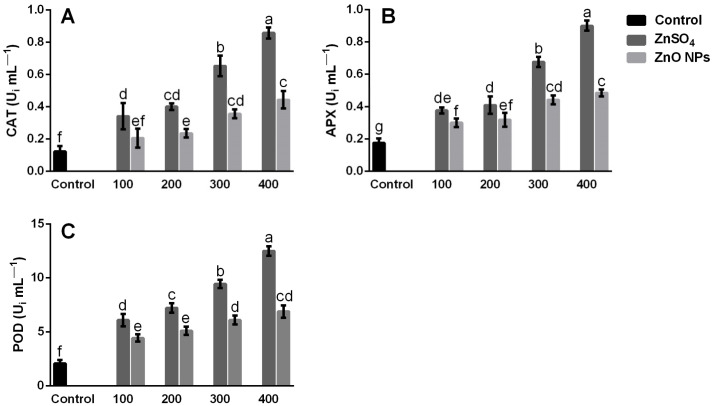
Activity of antioxidant enzymes in shoots of coriander plants grown for 58 days in inert soil modified with zinc sulfate (ZnSO_4_) and zinc oxide nanoparticles (ZnO NPs) at concentrations of 0, 100, 200, 300, and 400 mg of Zn/kg of soil. (**A**) Catalase activity (CAT), (**B**) Ascorbate peroxidase activity (APX), (**C**) Peroxidase activity (POD). Values are the average of five replications, means (n = 5). Bars represent the standard deviation of the mean. Different letters in each bar mean that the treatments were statistically different (Tukey, *p* ≤ 0.05).

## Data Availability

All the data incorporated in the manuscript.

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
