# Peer review of "Zinc Oxide Nanoparticles and Zinc Sulfate Impact Physiological Parameters and Boosts Lipid Peroxidation in Soil Grown Coriander Plants (Coriandrum sativum)"

_molecules, 2021, doi:10.3390/molecules26071998_

Round 1

Reviewer 1 Report

Dear authors

Manuscript Number: molecules-1152569. Title: Zinc Oxide Nanoparticles and Zinc Sulfate Impact Physiological Parameters and Boosts Lipid Peroxidation in Soil Grown Coriander Plants (Coriandrum sativum)

This is an interesting paper devoted to studying the physiological and antioxidant effect of Zn NPs and ZnSO4 on coriander plants.

Some topics have to be addressed before publication.

Introduction

Row 66…check letter size.

Row 76…It is suggested to finish a sentence after references in this line.

Row 86 to 97…please, prefer to build shorter sentences for better text comprehension.

Materials

Row 101 to 106… Morphology analysis by SEM is a methodology that must be separated from reagents and material suppliers.

Row 183 to 187… please, prefer to build shorter sentences for better text comprehension.

Results and Discussion

 In Figure 2.. the increase in carotenoid content would mean that plants are greener?...did they change in color during the experiment? Please comment on that.

Figure 2. … It is suggested that letters coming from statistical analysis appear over each bar and not inside them, in the same manner as they appear in the other Figures.

How could you explain the strange behavior in MDA results in Figure 4B, which rise and then decrease at higher concentration only in Zn2+ treatment?

It would be interesting to add some data regarding plant morphology (plant and shoot size) to support the authors' phytotoxicity conclusion. Is there any picture of the experiment?

Do the soil pH and conductivity change during the experiment? Did you measure any of these parameters to support better Zn 2+ dissolution and plant absorption?

There is a good description of analysis results and also a comparison of their results with other papers. However, a more in-depth discussion from the plant physiology perspective will improve this paper's quality.

 General recommendation.

It is strongly suggested that the authors review the whole document carefully and write paragraphs containing shorter sentences to follow the idea straightforwardly.

Author Response

March 21, 2021

Prof. Dr. Farid Chemat

Editor-in-Chief

Molecules - MDPI

Dear Dr. Chemat,

Per your request, we are submitting the revised version of our manuscript entitled “Zinc Oxide Nanoparticles and Zinc Sulfate Impact Physiological Parameters and Boosts Lipid Peroxidation in Soil Grown Coriander Plants (Coriandrum sativum)” Manuscript ID: molecules-1152569. We greatly appreciate the opportunity to respond and, where appropriate, we have changed the manuscript in accordance with the reviewers’ comments. The original comments are italicized, and the responses are indented and in regular font.

Reply: We thank the editor and Reviewers for the valuable time dedicated to evaluating our manuscript. We have improved it following the Reviewers’ comments. Reviewers have indicated that the english language and style are fine, and a minor spell check is required. In the corrected version of the manuscript, a language spell check was performed throughout the document. All changes were signaled using the "Track Changes" function in Microsoft Word, so that changes are easily visible to editors and reviewers. We believe the quality of the revised has been greatly improved and it is now ready to be published in Molecules.

Reviewer: 1

Comments

Manuscript Number: molecules-1152569. Title: Zinc Oxide Nanoparticles and Zinc Sulfate Impact Physiological Parameters and Boosts Lipid Peroxidation in Soil Grown Coriander Plants (Coriandrum sativum)

This is an interesting paper devoted to studying the physiological and antioxidant effect of Zn NPs and ZnSO4 on coriander plants.

Some topics have to be addressed before publication.

Reply: We thank the reviewer 1 for the time devoted to reviewing the document. We have addressed all the comments, and the answers are shown after each comment. We hope that the reviewer will find the revised version of our manuscript suitable for publication in Molecules.

  1. Page 2, Line 66: check letter size

Reply: We thank reviewer 1 for the correction. The font size was revised throughout the document, and is in accordance with the journal's guidelines.

  1. Page 2, Line 76. It is suggested to finish a sentence after references in this line. 

Reply: We appreciate reviewer 1 for this valuable comment, we agree with this suggestion. It now reads: Zinc (Zn) is an essential micronutrient involved in many cellular processes, includ-ing auxin biosynthesis, photosynthesis, and protein synthesis [19,20]. Nevertheless, the concentration of Zn in the soil is very low, and it occurs in the form of various types of salts, including sphalerite (ZnFe)S, zincite (ZnO) and smithsonite ZnCO3. Furthermore, the absorption of this element by plants is determined by the concentration of carbonates (CaCO3) and soil pH, which are the main causes that limit the availability of this micro-nutrient [2,21].

  1. Page 3, Line 86-98: Please, prefer to build shorter sentences for better text comprehension. 

Reply: We thank reviewer 1 for these comments, short sentences were constructed for a better understanding of the text. The paragraph now reads: Several studies have addressed the impact of ZnO NPs on physiological and stress parameters in plants [7,8,23]. However, to verify the feasibility of ZnO NPs as a fertilizer in plant nutrition, it is necessary to carry out simultaneous comparative studies to determine the efficiency of nanofertilizers (ZnO NPs) and conventional fertilizers (ZnSO4).

Thus, it is important to determine the relative degrees of importance of the release of ions and its absorption in plant tissues, in addition to physiological and stress impacts, which can compromise plants development.

Therefore, the objective of this study was to evaluate the exposure effects of ZnSO4 and ZnO NPs in coriander plants at concentrations of 0, 100, 200, 300, and 400 mg of Zn/kg of soil. The accumulation of Zn in the tissues, the content of photosynthetic pigments, the responses of oxidative stress parameters and the activity of antioxidant enzymes were determined to compare the effects of ZnO NPs with other Zn species (ZnSO4) to elucidate mechanisms of toxicity.

  1. Page 4, Line 101-106: Morphology analysis by SEM is a methodology that must be separated from reagents and material suppliers.

We agree with reviewer 1 in this comment, section 2.1.2 was generated, with the subtitle “Characteristics of the ZnO NPs Used in This Experiment”. In which only the methodology used for the characterization of the ZnO NPs is described.

New section now reads;

        2.1.2. Characteristics of the ZnO NPs Used in this Experiment

        The morphology and microstructure of the samples were examined by conventional and                 high-resolution transmission electron microscopy (TEM and HRTEM), and selected area         electron diffraction (SAED) using a FEI-TITAN 80-300kV microscope (Fisher Scientific,          Hillsboro, OR), operated at an acceleration voltage of 300 kV. TEM and HRTEM         micrographs were processed using a Fast Fourier transform software (Digital Micrograph               3.7.0, Gatan Software, Pleasanton, CA).

  1. Page 4, Line 183-187: Please, prefer to build shorter sentences for better text comprehension.

We appreciate reviewer 1 for this valuable comment. Thus, for the best understanding of this section, shorter sentences were made. The paragraph now reads: The MDA content in the shoots was determined according to Wang et al. [30]. The plant fresh tissue (0.3 g) was homogenized in 3 mL of 10% TCA, the mixture was subjected to centrifugation for 10 min at 12,000 rpm. Two mL of the supernatant were recovered and added to 4 mL of 0.6% thiobarbituric acid (TBA, in 10% TCA), and incubated at 100 °C in a water bath for 15 min.

  1. Page 8: In Figure 2. the increase in carotenoid content would mean that plants are greener?...did they change in color during the experiment? Please comment on that.

Reply: We thank reviewer 1 for this helpful comment. Thus, for a better understanding of this section, the following section has been included: The intensity of the green color of the leaves is due to the presence of photosynthetic pigments (chlorophyll). In the photosynthesis process, carotenoids help to capture light, but they also have an important function as they are considered the first line of defense against the accumulation of H2O2, which is an integral part of ROS that generate oxidative stress due to the absorption of metals heavy on plants [39,40]. Therefore, the increases in chlorophyll and carotenoids in coriander plants exposed to the different concentrations and sources of Zn, may be related to the greater increase in H2O2 (Figure 4). The production of excessive ROS can inhibit photosynthetic processes and, consequently, cause a greater absorption of light energy that can be used during photosynthesis [41].

Additional comment: during the development of the experiment, the impact of the Zn sources on the chlorophyll and carotenoid content could not be determined, because the methodology used for the quantification of photosynthetic pigments is a destructive method.

  1. Figure 2. It is suggested that letters coming from statistical analysis appear over each bar and not inside them, in the same manner as they appear in the other Figures. 

Reply: We thank reviewer 1 for this observation. After the reviewer's comment, we have corrected figure 3, and it can be seen that the letters are over each bar.

  1. How could you explain the strange behavior in MDA results in Figure 4B, which rise and then decrease at higher concentration only in Zn2+ treatment?. 

Reply: We thank the reviewer for the questioning, to give a better explanation to the MDA results, the following section was incorporated: The accumulation of MDA is a byproduct of oxidative damage to membrane lipids. In coriander plants subjected to different sources and concentrations of Zn, it is indicative of an increase in lipid peroxidation and is in accordance with the observation reported by several authors in different plant species [7,12]. In plant cells, redox homeostasis is developed because of the equilibrium between ROS generation and the functioning of the antioxidant enzymes where an efficient defense system in plants keeps the proper balance between ROS generation and elimination [48]. A basal level of ROS, which is maintained above the toxic concentration, redox regulations in the cells are essential since they allow to maintain the balance between the production of ROS and their elimination which can occur through the activity antioxidant enzymes and uphold the usual physiological activities of plants [45].

  1. It would be interesting to add some data regarding plant morphology (plant and shoot size) to support the authors' phytotoxicity conclusion. Is there any picture of the experiment?.

 Reply: We thank the reviewer 1 for the valuable time spent reading and making constructive comments in this section. However, all the research information has been presented in the manuscript that was sent for review.

  1. Do the soil pH and conductivity change during the experiment? Did you measure any of these parameters to support better Zn 2+ dissolution and plant absorption?. 

Reply: We thank the reviewer 1 for these observations. The pH and conductivity were only determined in the solutions with the fertilization of macronutrients before their application. We consider that the elemental determination of Zn by ICP-AES allows to corroborate a greater absorption of Zn in plants due to the ionization of the element, results that are in agreement with the observations reported by several authors in different plant species. References have been cited in the article.

  1. There is a good description of analysis results and also a comparison of their results with other papers. However, a more in-depth discussion from the plant physiology perspective will improve this paper's quality. 

Reply: We thank reviewer 1 for his valuable comments and observations, which were considered for the corrected version of the manuscript. Throughout the manuscript, new discussions were incorporated that deepen the impact of the results on plant physiology and biochemistry.

  1. It is strongly suggested that the authors review the whole document carefully and write paragraphs containing shorter sentences to follow the idea straightforwardly..

 Reply: We thank reviewer 1 for the valuable time spent reading and making constructive comments on this document. We have followed all the comments, and the corresponding responses are shown after each comment. We hope the reviewer will find the revised version of our manuscript appropriate for publication in Molecules.

Reviewer 2 Report

Review of the article "Zinc Oxide Nanoparticles and Zinc Sulfate Impact Physiologi-2 cal Parameters and Boosts Lipid Peroxidation in Soil Grown 3 Coriander Plants (Coriandrum sativum)". The article is devoted to the influence of different forms of Zn on coriander plants. In general, the article seemed interesting to me, it is very easy to read, however, the article contains two significant points regarding the conclusions from the work and one point related to the methodology.
Major points
- What exactly did we learn new from this article? This needs to be formulated in several sentences, clearly and understandably! It is not at all clear whether there is stress. On the one hand, plants have more photosynthetic pigments, which is an indirect sign of faster assimilation, which means a potentially greater yield. On the other hand, the plants that received Zn have an increased concentration of hydrogen peroxide and MDA, and more antioxidant enzymes. So is there stress, perhaps the plant has embarked on the path of greater productivity and, due to the acceleration of metabolism, is there a redox imbalance?
- The question of the productivity of Zn in plants during experiments on soil is always highly ambiguous. Investigators try to calcine or disinfect the soil in laboratory conditions, but it is very quickly colonized by microflora. It is known that ZnO nanoparticles are powerful antibacterial agents. Yesterday there was an excellent review article on this matter (https://doi.org/10.3389/fphy.2021.641481). It is found out, ZnO nanoparticles can significantly affect both pathogenic and obligate microflora. At least part of the effects of ZnO nanoparticles in plants grown on soil is associated with the effect on soil bacteria communities. This point should be taken into account by authors in a future version of the manuscript.
- The concentration of hydrogen peroxide in the plant, according to the authors, is of the order of 1 mM (34 μg/g). This is an extremely high concentration of hydrogen peroxide corresponding to the terminal stage of oxidative stress. The authors measure the concentration by titration of KJ. In plant extracts, KJ reacts with a large number of compounds. And not only not in plant extracts. KJ in the light is oxidized even by molecular oxygen in the air, turning yellow due to the liberated free iodine. In general, authors need to either double-check their data using a more applicable method, or replace the units with relative units.
Minor points
- No bar size on TEM photo. The size of the bar must be applied.
- Copper grids were used for microscopy. Why? Can copper potentially interact with zinc oxide nanoparticles?
- (U, ml-1) - what does this unit mean, perhaps you need U per ml or U * ml-1 or U / ml (Fig 5).

Author Response

March 21, 2021

Prof. Dr. Farid Chemat

Editor-in-Chief

Molecules - MDPI

Dear Dr. Chemat,

Per your request, we are submitting the revised version of our manuscript entitled “Zinc Oxide Nanoparticles and Zinc Sulfate Impact Physiological Parameters and Boosts Lipid Peroxidation in Soil Grown Coriander Plants (Coriandrum sativum)” Manuscript ID: molecules-1152569. We greatly appreciate the opportunity to respond and, where appropriate, we have changed the manuscript in accordance with the reviewers’ comments. The original comments are italicized, and the responses are indented and in regular font.

Reply: We thank the editor and Reviewers for the valuable time dedicated to evaluating our manuscript. We have improved it following the Reviewers’ comments. Reviewers have indicated that the english language and style are fine, and a minor spell check is required. In the corrected version of the manuscript, a language spell check was performed throughout the document. All changes were signaled using the "Track Changes" function in Microsoft Word, so that changes are easily visible to editors and reviewers. We believe the quality of the revised has been greatly improved and it is now ready to be published in Molecules.

Reviewer: 2

Comments

Review of the article "Zinc Oxide Nanoparticles and Zinc Sulfate Impact Physiologi-2 cal Parameters and Boosts Lipid Peroxidation in Soil Grown Coriander Plants (Coriandrum sativum)". The article is devoted to the influence of different forms of Zn on coriander plants. In general, the article seemed interesting to me, it is very easy to read, however, the article contains two significant points regarding the conclusions from the work and one point related to the methodology.

Reply: We thank the reviewer 2 for the valuable time spent reading and making constructive comments on this manuscript. In addition, we thank reviewer 2 for considering the research work interesting. We have followed all comments and the corresponding responses are displayed after each comment. We hope that the reviewer will find the revised version of our manuscript suitable for publication in Molecules.

  1. What exactly did we learn new from this article? This needs to be formulated in several sentences, clearly and understandably!

Reply: We thank reviewer 2 for his suggestions and constructive comments. We consider that the physiological and biochemical impacts of the different forms of Zn suggest that the Zn ions derived from the ZnO NPs exert preferential toxicity in plants. Therefore, this source of Zn can be considered as a safe fertilizer for its application in biological species. These statements have been pointed out throughout the manuscript, and are part of the knowledge generation of this research.

  1. It is not at all clear whether there is stress. On the one hand, plants have more photosynthetic pigments, which is an indirect sign of faster assimilation, which means a potentially greater yield. On the other hand, the plants that received Zn have an increased concentration of hydrogen peroxide and MDA, and more antioxidant enzymes. So is there stress, perhaps the plant has embarked on the path of greater productivity and, due to the acceleration of metabolism, is there a redox imbalance?.

Reply: We thank reviewer 2 for these constructive suggestions to improve the content of this article. In addition, following the reviewer's suggestions, the following section was written to better describe and understand the stress section: It has been recognized that the increase in H2O2 and MDA by the absorption of heavy metals are a byproduct of the oxidation of polyunsaturated lipids caused by ROS species [7,12]. Low concentrations of H2O2 (10 μm) can inhibit CO2 fixation by 50% and therefore, the accumulation of photosynthetic pigments [44].

Therefore, the balance between ROS production and elimination in chloroplasts is delicate and must be strictly controlled by the activity of antioxidant enzymes and components of secondary metabolites [44,45]. This is perhaps surprising, however, if it is considered that ROS generated within the cell participate in a redox signaling cascade, then could be related to maintain an optimal level of ROS at the cellular level that allows adequate redox biological reactions and the regulation of the numerous essential processes for plants such as growth and development [46,47].

The data from this study suggest that the higher activity of antioxidant enzymes (CAT, APX and POD, Figure 5), could participate in the modulation of ROS, since the levels of H2O2 (Figure 4) in coriander plants are not present in quantities that significantly affect the accumulation of photosynthetic pigments (Figure 3). This indicates that the stress levels generated by the application of Zn in its different sources and concentrations were modulated by the activity of antioxidant enzymes, and did not affect the chlorophyll content, which is a common indicator of photosynthetic efficiency and one of the most important determinants of plant growth.

  1. The question of the productivity of Zn in plants during experiments on soil is always highly ambiguous. Investigators try to calcine or disinfect the soil in laboratory conditions, but it is very quickly colonized by microflora. It is known that ZnO nanoparticles are powerful antibacterial agents. Yesterday there was an excellent review article on this matter (https://doi.org/10.3389/fphy.2021.641481). It is found out, ZnO nanoparticles can significantly affect both pathogenic and obligate microflora. At least part of the effects of ZnO nanoparticles in plants grown on soil is associated with the effect on soil bacteria communities. This point should be taken into account by authors in a future version of the manuscript.

Reply: We thank reviewer 2 for these constructive suggestions to improve the content of this article. The suggested article (https://doi.org/10.3389/fphy.2021.641481) was revised, and it was very useful to reinforce the introduction of the manuscript (lines 49 to 52) and the explanation of the hormesis process in the section on antioxidant enzymes (Lines 477 to 487).

The new sections are now read:

Introduction (lines 49-52)

In biology and medicine, the possibility of using ZnO NPs is considered for their cytostatic activity against cancer cells, the antimicrobial and fungicidal activity because of their high antibacterial efficacy at low concentrations, and their activity against a wide range of strains [6].

Results and Discussion (477-487)

A high concentration of ROS stimulates the activity of antioxidant enzymes [14]. Antioxidant enzymes play an important role in protecting cellular components from oxidative damage caused by the absorption of heavy metals in plants [52]. The response of defense reactions against ROS is called hormesis [53], which induces defense mechanisms on two levels. The first is the enzymatic level (short-term reaction), in which the activity of antioxidant enzymes is increased. The second level is long-term adaptation, and comprises two sublevels: transcriptional and genomic [6]. Plants have differential responses to oxidative stress conditions, which allow them to develop resistance to heavy metal exposure [45].

Additional comment: In this study, inert soil with neutral pH and without mineral nutrient content was used, since the only sources of variation were zinc sources (ZnSO4 and ZnO NPs) at different concentrations. The above in order to avoid the influence of sources of variation that were not considered within the study (Confusing factors). On the other hand, we did not consider making inferences about the effect of soil bacterial communities on plants due to the influence of ZnO NPs, since soil bacterial communities were not considered as a source of variation in the study. However, the article (https://doi.org/10.3389/fphy.2021.641481), helped us to reinforce the sections in the document that were previously mentioned.

  1. The concentration of hydrogen peroxide in the plant, according to the authors, is of the order of 1 mM (34 μg/g). This is an extremely high concentration of hydrogen peroxide corresponding to the terminal stage of oxidative stress. The authors measure the concentration by titration of KJ. In plant extracts, KJ reacts with a large number of compounds. And not only not in plant extracts. KJ in the light is oxidized even by molecular oxygen in the air, turning yellow due to the liberated free iodine. In general, authors need to either double-check their data using a more applicable method, or replace the units with relative units.

Reply: We thank reviewer 2 for his constructive comment and suggestions. Due to the methodologies applied during the development of this research, we consider that there may be interference with some compounds that have a similar absorption wavelength. However, the extraction carried out was directed to the compound of interest (H2O2), in addition, the extraction and quantification was carried out based on that established by other investigations that have already validated this methodology (references have been cited in the article). On the other hand, the results show a trend in the relationship between the dependent variable (H2O2) and the independent variable (treatments with Zn), with a low standard deviation, which helps to improve the precision of the analysis. The foregoing allows us to make inferences and/or conclusions that are within the objectives established in this research.

  1. No bar size on TEM photo. The size of the bar must be applied.

Reply: We thank reviewer 2 for the observation. In the lower left part of the micrograph (Figure 1) the scale is presented, which corresponds to 200 nm. In the same way, the presented histogram supports the information obtained by the micrograph.

  1. Copper grids were used for microscopy. Why? Can copper potentially interact with zinc oxide nanoparticles?

Reply: We thank reviewer 2 for these questions. Since the samples analyzed by the TEM equipment must be a few millimeters thick, a grid is required to support the sample. There are different types of grids, however, the copper grid is the most suitable for the analysis of metallic nanoparticles.

References

Tizro P., Choi C., Khanlou N. (2019) Sample Preparation for Transmission Electron Microscopy. In: Yong W. (eds) Biobanking. Methods in Molecular Biology, vol 1897. Humana Press, New York, NY. https://doi.org/10.1007/978-1-4939-8935-5_33.

Sample Preparation Techniques in Analytical Chemistry, Edited by Somenath Mitra ISBN 0-471-32845-6 Copyright 6 2003 John Wiley & Sons, Inc.

  1. - (U, ml-1) - what does this unit mean, perhaps you need U per ml or U * ml-1 or U / ml (Fig 5).

Reply: We thank reviewer 2 for the observation. The meaning of the unit has already been presented in lines 234-237. New section now reads: Enzyme activity was expressed in units per volume (Ui mL-1), one Ui being the amount of enzyme that transforms one micromole of substrate per minute under the standard conditions mentioned above.

  1. Can copper potentially interact with zinc oxide nanoparticles.

Reply: According to the periodic properties of metals, the chemical reactivity of copper (Cu) is lower than that of zinc (Zn), therefore, this metal does not interact chemically with the nanoparticles.

We highly appreciate the suggestions of the reviewers, which greatly contributed to the improvement of our manuscript.

Round 2

Reviewer 2 Report

The authors have significantly improved the text of the manuscript